# Associated Lichen Sclerosis Increases the Risk of Lymph Node Metastases of Vulvar Cancer

**DOI:** 10.3390/jcm9010250

**Published:** 2020-01-17

**Authors:** Yohann Dabi, Marie Gosset, Sylvie Bastuji-Garin, Rana Mitri-Frangieh, Sofiane Bendifallah, Emile Darai, Bernard Jean Paniel, Roman Rouzier, Bassam Haddad, Cyril Touboul

**Affiliations:** 1Faculté de médecine de Créteil UPEC—Paris XII. Service de Gynécologie-Obstétrique et Médecine de la Reproduction. Centre Hospitalier Intercommunal de Créteil. 40 Avenue de Verdun, 94000 Créteil, France; yohann.dabi@gmail.com (Y.D.); marie-gosset@wanadoo.fr (M.G.); bernard-jean.paniel@chicreteil.fr (B.J.P.); bassam.haddad@chicreteil.fr (B.H.); 2Université Paris Est, A-TVB DHU, LIC EA4393, 94010 Créteil, France; sylvie.bastuji-garin@hmn.aphp.fr; 3AP-HP, Hôpital Henri Mondor, Service de Santé Publique, 94010 Créteil, France; 4AP-HP, Hôpital Henri Mondor, Unité de Recherche Clinique (URC Mondor), 94010 Créteil, France; 5Service d’anatomopathologie, Centre Hospitalier Intercommunal de Créteil. 40 Avenue de Verdun, 94000 Créteil, France; rana.mitri-frangieh@chicreteil.fr; 6Service de Gynécologie-Obstétrique et Médecine de la Reproduction, Hôpital Tenon (AP-HP), 4 rue de la Chine, 75020 Paris, France; sofiane.bendifallah@yahoo.fr (S.B.); emile.darai@aphp.fr (E.D.); 7Department of Surgery, Institut Curie, Paris, Saint-Cloud, France; EA 7285; Université Versailles-Saint-Quentin-en-Yvelines, 35 Rue Dailly, 92210 Saint-Cloud, France; roman.rouzier@yahoo.fr

**Keywords:** vulvar cancer, lichen sclerosis, lymph node involvement, prognostic factors, vulvar surgery, survival

## Abstract

The most important prognostic factor in vulvar cancer is inguinal lymph node status at the time of diagnosis, even in locally advanced vulvar tumors. The aim of our study was to identify the risk factors of lymph node involvement in these women, especially the impact of lichen sclerosis (LS). We conducted a retrospective population-based cross-sectional study in two French referral gynecologic oncology institutions. We included all women diagnosed with a primary invasive vulvar cancer. Epithelial alteration adjacent to the invasive carcinoma was found in 96.8% (*n* = 395). The most frequently associated was LS in 27.7% (*n* = 113). In univariate analysis, LS (*p* = 0.009); usual type VIN (*p* = 0.04); tumor size >2 cm and/or local extension to vagina, urethra or anus (*p* < 0.01), positive margins (*p* < 0.01), thickness (*p* < 0.01) and lymphovascular space invasion (LVSI) (*p* < 0.01) were significantly associated with lymph node involvement. In multivariate analysis, only LS (OR 2.3, 95% CI [1.2–4.3]) and LVSI (OR 5.6, 95% CI [1.7–18.6]) remained significantly associated with positive lymph node. LS was significantly associated with older patients (*p* = 0.005), anterior localization (*p* = 0.017) and local extension (tumor size > 2 cm: *p* = 0.001). LS surrounding vulvar cancer is an independent factor of lymph node involvement, with local extension and LVSI.

## 1. Introduction

Vulvar cancer is a rare disease representing only 0.3% of all cancers in women and from 3% to 5% of gynecologic cancers [1,2]. It is most frequently seen in women between 65 and 75 years [3,4]. Ninety percent of vulvar cancers are squamous cell carcinoma (SCC), whereas the other histologic subtypes such as melanoma, Paget’s disease and adenocarcinoma of the Bartholin gland are less frequent [5]. 

The most important prognostic factor is inguinal lymph node status at the time of diagnosis [6,7,8], even in locally advanced vulvar tumors: The 5-year and 10-year survival of patients without metastatic nodes are 62% and 47% respectively, versus 39% and 27% for patients with metastatic nodes [7]. Besides the classic predictive factors of inguinal lymph node involvement, such as depth of invasion and tumor size, the influence of the type of surrounding epithelial disorders may be important. Epithelial disorders are found in more than 80% of patients. They are subdivided into lesions due to carcinogenic genotypes of human papillomavirus (HPV) including unifocal and multifocal usual vulvar intraepithelial neoplasia (uVIN) and non HPV lesions in which lichen sclerosis (LS) is the most frequent observed [9,10,11,12]. It is known that the type of epithelial disorder has prognostic significance in patients with vulvar cancer [9,10,11,12,13], but there are few data to date about clinical and prognostic features of patients with LS.

The objective of this study was to identify the risk factors of lymph node involvement in women with vulvar cancer and especially the impact of LS. 

## 2. Materials and Methods

The study protocol was approved by the Ethics Committee of Paris X, France. The present report complies with the Strengthening the Reporting of Observational Studies in Epidemiology (STROBE) statement [14].

Our departments are referral centers for vulvar disease. We treat about 25 vulvar cancers per year. For this cross-sectional analysis, we examined the records of consecutive patients with invasive vulvar carcinoma managed in the tertiary University Hospital of Creteil from 1992 to 2012 and the University Tenon Hospital (AP-HP) from 2005 to 2010. We excluded patients with recurrent vulvar carcinoma or in situ carcinoma. The following data were recorded: age, body mass index (BMI), American Society of Anesthesiologists (ASA) status, site of lesion, histologic type and characteristics, surgical treatment, complications, adjuvant treatment, relapse and survival data. Stages were assigned based on clinical data and lymph node pathology when available according to the 2009 FIGO (International Federation of Gynecology and Obstetrics) staging [6]. Patients who had been staged before 2009 were retrospectively assigned for the data collection. Surgical treatment of the vulva was radical and conservative (hemivulvectomy or local excision) when the surgical excision could encompass the lesion with at least a 1-cm margin of clinically normal skin. Patients with exclusive lateral vulvar lesions had a unilateral lymphadenectomy. Groin node dissection was avoided in patients older than 70 years without clinically metastatic nodes, because of their medical status or in cases of superficially invasive carcinoma (<1 mm). Type of dissection was reviewed according to the classification described by Rouzier et al. [15]. Since 2002, sentinel node identification was progressively performed in our center, consistent with the guidelines from the literature and the Collège National des Gynécologues et Obstétriciens Français (CNGOF) [16,17,18,19]. Patients with invasive vulvar cancer more than 1 mm, of clinical stages T1 and T2 underwent sentinel lymph node detection, bilateral in case of midline or close-to-midline lesions (<1 cm from the midline). The presence of clinically suspicious or palpable lymph nodes was a contraindication to performing the sentinel node procedure [16,17,18,19]. The technic used for sentinel lymph node mapping was the one described by Christine Louis Sylvestre et al. [19] using a double detection method.

Adjuvant therapy decision was based on final pathological analysis, and both centers followed latest ESGO guidelines at the time the patient was treated.

The associated epithelial disorders were retrospectively separated according to the International Society for the Study of Vulvar Disease (ISSVD) [10,13,20,21]. Because most of the data were not interpretable with the last ISSVD classification, we used the previous ISSVD classification [22].

### Statistical Analysis

Characteristics of the included women are described as a number (%) for qualitative variables and median (interquartile range, IQR) for quantitative variables and compared according to presence of LS. To identify factors associated with a poorer prognosis, we compared initial characteristics of patients with and without lymph node involvement using the chi-squared test or Fisher’s exact test, as appropriate, for categorical variables and the nonparametric Kruskal–Wallis test for continuous variables. Odds ratios (ORs) and 95% confidence intervals (CIs) were estimated using univariate asymptotic or exact logistic regression analyses, as appropriate, for variables that yielded *p* values < 0.05. These variables were selected for multivariate analyses. As we were interested in LS, we did not include the variable “usual type VIN” in the multivariate analysis. Confounders and interactions were tested in bivariate models.

All tests were two-sided, and *p* values ≤ 0.05 were considered significant. Analyses were performed using STATA software version 12.0 (StataCorp, College Station, TX, USA) and the R software (http://cran.r-project.org).

## 3. Results

Four hundred eight patients with invasive vulvar carcinoma were included. The median age of the population was 71.8 years (interquartile range (IQR) 58.1–79.1). The median BMI was 24.0 (IQR 21.2–28.0) (Table 1). Among the 408 patients, 94.9% (*n* = 387) had a squamous cell carcinoma (SCC), 2.2% (*n* = 9) had an adenocarcinoma and 2.9% (*n* = 12) another type such as melanoma. 

Epithelial alteration adjacent to the invasive carcinoma was found in 96.8% (*n* = 395). The most frequent associated epithelial disorder was lichen, LS in 27.7% (*n* = 113) and squamous cell hyperplasia in 14.7% (*n* = 60). Usual VIN (HPV induced) was found in 31.4% (*n* = 128), with 22.5% of unifocal classic VIN and 9.9% of multifocal extensive VIN. Thirteen percent of patients with LS were treated by local corticoids. The median stromal invasion was 3 mm (IQR 1–9).

The most frequent localization was labia minora (33.8%, *n* = 138) and 16.6% (*n* = 68) had a rectum or vaginal extension (Table 1). Thirty seven percent of patients (*n* = 140) had stage I cancer, 25.7% (*n* = 96) stage II, 26.7% (*n* = 100) stage III and 10.2% (*n* = 38) stage IV (34 missing data). Patients had positive lymph nodes in 31.9% (*n* = 130).

Surgery consisted of a total vulvectomy in 45.1% (*n* = 184), anterior vulvectomy in 17.2% (*n* = 70), posterior vulvectomy in 9.8% (*n* = 40), lateral vulvectomy in 12.7% (*n* = 52) and a partial vulvectomy in 13.5% (*n* = 55). Despite attempting to resect 1 cm normal skin margins around the tumor, in case of tumor located near the urethra or the anus, 22.5% patients had histologic positive margins (*n* = 92). Lymphadenectomy was unilateral in 18.9% (*n* = 77), bilateral in 68.1% (*n* = 278), and 13.0% had no groin dissection (*n* = 53) because of their medical status, an age greater than 70 years without clinically metastatic nodes or in cases of superficially invasive carcinoma (<1 mm). In line with Rouzier’s classification [15], 37.3% (*n* = 152) underwent an inguinal and medial femoral lymphadenectomy, 19.9% an inguinofemoral dissection (*n* = 81, 64 of which involved ligature of the saphenous vein), 4.4% (*n* = 18) a superficial inguinal dissection and 6.4% (*n* = 26) a sentinel lymph node procedure. 

Median hospitalization was 11 days (IQR 5–16). Short-term post-operative complications were frequent: 16.2% patients (*n* = 66) had an infection, 20.6% (*n* = 84) a wound dehiscence, 10.8% (*n* = 44) a lymphocele and 3.4% (*n* = 14) a phlebitis. 

Radiotherapy was delivered to the inguinal area in 17.4% patients (*n* = 71). 

### 3.1. Factors Associated with Lymph Node Involvement

In univariate analysis, the following variables were significantly associated with lymph node involvement: the presence of LS (OR 1.89, 95% CI [1.15–3.13], *p* = 0.009); tumor size > 2 cm (OR 2.11, 95% CI [1.32–3.41], *p* = 0.001) and/or local extension to vagina, urethra or anus (OR 6.02, 95% CI [2.86–12.66], *p* = 0.001); thickness (*p* < 0.0001); positive margins (OR 0.81, 95% CI [0.44–1.45], *p* < 0.01) and lymphovascular space invasion (LVSI) (OR 5.16, 95% CI [1.70–18.81], *p* < 0.01) (Table 2). As usual VIN was a protective factor (OR 0.38 95% CI [0.33–0.98], *p* = 0.04), we did not include this variable in the multivariable analysis. In multivariate analysis, only presence of LS (OR 2.3, 95% CI [1.2–4.3], *p* = 0.01) and LVSI (OR 5.6, 95% CI [1.7–18.6], *p* = 0.05) were significantly associated with lymph node involvement (Table 2).

### 3.2. Characteristics of the Population with Lichen Sclerosis

Because the presence of LS had a strong and independent predictive value for positive lymph node status, we analyzed patients’ characteristics associated with this epithelial disorder (Table 3). These patients differed significantly from the global population. LS was significantly associated with older patients (*p* = 0.005), anterior localization (*p* = 0.017), tumor size > 2 cm (*p* = 0.0005), positive margins (*p* = 0.03) and thickness (*p* < 0.0001). We found an OR for the risk of lymph node involvement of 1.89 [1.14–3.13]. To the contrary, the absence of LS was significantly associated with FIGO 1 (*p* = 0.009) and tumor size < 2 cm (*p* = 0.004).

## 4. Discussion

### 4.1. Main Findings

We found that the type of epithelial disorder associated with invasive vulvar carcinoma was an important prognostic factor of lymph node involvement. While LS was significantly associated with positive lymph node (38.9% versus 25.1% in the absence of LS, *p* = 0.009), uVIN emerged as a protective factor (23.8% versus 35.1% in the absence of uVIN, *p* = 0.04). In multivariate analysis, LS remained an independent risk factor, as well as LVSI. 

### 4.2. Strengths and Limitations

A strength of our study is that the results are based on a large cohort (more than 400 patients) since vulvar cancer is a rare disease. We were able to establish a significant correlation between simple clinical or histologic characteristics and lymph node involvement, which is a major prognostic factor. However, it was a retrospective, two-center study spanning a period of 20 years. During this period, the FIGO staging system was revised to reflect the importance of lymph node involvement as a prognostic factor of vulvar cancer [6,7,8]. Four major changes were thus introduced in 2009 to further characterize stage III, which previously comprised a heterogeneous group of patients with negative or positive nodes. To take this into account, patients who had been staged before 2009 were retrospectively assigned for data collection, rendering analysis more difficult. Another change involved the classification of VIN, which was modified by the ISSVD in 2004. Prior to this date, dVIN was not clearly identified and we chose to use the previous classification [22] as most of the data are older than 2004. This is a limit of the present study and would need a review of the histologic slides. But LS is a clinical entity, and the significative association of LS with lymph node involvement, independently of the presence of atypia, softens this limit. Eventually, since this information was not recorded in our databases, we cannot confirm or infirm that the adjuvant therapy was performed according to the same protocol (especially radiotherapy), which might have introduced a bias.

### 4.3. Interpretation

According to previous reports [10,12,13,20], epithelial disorders associated with vulvar carcinoma are divided into two distinct etiologies: LS and HPV induced disorders, respectively associated with dVIN and uVIN (Table 4). Usual VIN is HPV related in most cases, including warty, basaloid and mixed type. It is typically seen as a unifocal or multifocal lesion with a variety of clinical presentations: erosions, plaques and nodules, pigmented, red or white. Invasive squamous carcinomas of warty or basaloid type are associated with uVIN. To the contrary, differentiated VIN is seen particularly in older women with LS and/or squamous cell hyperplasia in some cases [12]. Neither dVIN, nor associated keratinizing SSC is HPV related [10,11,20]. LS was present in 27.7% of our patients, in accordance with another large study of 1287 patients with vulvar carcinoma reporting 33% of patients with LS [23]. 

The prognostic significance of adjacent lesions in the setting of vulvar cancer has already been studied. Before the new classification of epithelial disorders proposed by the ISSVD in 2004 [10], the presence of VIN in adjacent lesions had been identified as a predictor of reduced disease-free survival, even before using the recent classification of epithelial disorders [24]. However, in 2001, Rouzier et al. showed that the presence of HPV-related VIN was a positive prognostic factor [9]. In their article, patients with uVIN had a 5-year survival rate of 87%, against 42% for patients with non-neoplastic epithelial disorders, dVIN or without associated epithelial alteration (*p* < 0.01). However, the data regarding lymph node involvement was not reported. Our study shows that 30 patients (23.8%) with uVIN had positive lymph nodes. Usual VIN was more likely to be associated with superficially invasive lesions, which have an excellent prognosis: no spread to the groin has been reported for a depth of infiltration <1 mm [25,26,27]. Most of the authors studied HPV-related uVIN [9,24,28]. The good prognosis of HPV is widely proven and also well known in oropharyngeal cancers [29]. However, only a few studied the prognostic significance of LS in vulvar cancer, which is more frequent and underdiagnosed [12,30,31].

In our study, the site of onset in case of LS was more frequently of anterior localization and of larger and deeper extension. These differences in invasion may be due to a later diagnosis in women with LS. As they are older, they may be more prone to ignoring lesions than women with HPV-related VIN. Moreover, the clinical distinction with cancer can be difficult.

Another hypothesis is that these two forms of dermatosis could have a difference in invasive potential. Ansink et al. showed that patients with an HPV-positive pCR tumor had a better prognosis than those with an HPV-negative pCR tumor (*p* = 0.03) [28]. Histologically, HPV-independent vulvar carcinomas—associated with dVIN and LS—frequently show mutations of p53 and are histologically keratinizing, whereas HPV-associated vulvar carcinomas are of the basaloid or warty type and arise from uVIN. The viral gene of HPV is involved in a specific process of malignant transformation; p16 immunohistochemistry is diffusely positive in these lesions, and a high Ki-67 proliferation index is observed [13,32]. These two different oncogenic pathways lead to different degrees of malignancy. Indeed, our study suggests two different natural histories and locoregional power of invasion, depending on the type of adjacent epithelial disorder. In the literature, dVIN has a higher risk of progression to invasive SCC than uVIN (33% vs. 5.7%, respectively). In addition, time to progression to SCC is significantly shorter in women with dVIN compared with those with uVIN [13,33,34,35]. Consequently, it could be interesting to compare the prognosis of patients with and without lymphadenectomy and the effect of radiotherapy depending the type of VIN.

## 5. Conclusions

In conclusion, LS surrounding vulvar cancer is an independent predictive factor of positive lymph node status. LS has its own clinical presentation requiring careful monitoring, independently of the presence of atypia. A biopsy should be performed to identify invasive lesion as soon as a suspect area is observed as LS increases the risk of lymph nodes metastasis. Prospective clinical studies could show whether the treatment of this pre-invasive lesion could reduce the risk of vulvar carcinoma.

## Figures and Tables

**Table 1 jcm-09-00250-t001:** Clinicopathologic characteristic of the population (*n* = 408).

Clinicopathologic Characteristic of the Population	Overall (*n* = 408)
Age, years	71.8 [58.1–79.1]
Body mass index, Kg/m^2^	24 [21.2–28.0]
**Histologic type**	
Squamous cell carcinoma	387 (94.9%)
Adenocarcinoma	9 (2.2%)
Others	12 (2.9%)
**Localization**	
Anterior	254 (62.3%)
Detailed	
Labia	209 (51.2%)
Labia minora	138 (33.8%)
Labia majora	73 (17.9%)
Clitoris	108 (26.5%)
Urethra	19 (4.7%)
Vagina	56 (13.7%)
Perineum	79 (19.4%)
Rectum	12 (2.9%)
**FIGO classification**	
I	140 (37.4%)
II	96 (25.7%)
III	100 (26.7%)
IV	38 (10.2%)
**Local extension**	
Tumor confined to vulva, size < 2 cm	174 (43.2%)
Tumor confined to vulva, size > 2 cm	189 (46.9%)
Tumor size > 2 cm and/or extension to vagina, urethra or anus	40 (9.9%)
**Lymph node status**	
Positive lymph node	130 (31.9%)
Negative lymph node	230 (56.4%)
Unknown status	48 (11.7%)
**Epithelial disorder**	395 (96.8%)
Lichen sclerosis	113 (27.7%)
Squamous cell hyperplasia	60 (14.7%)
Paget disease	6 (1.5%)
Usual type VIN	
Unifocal usual type VIN	89 (21.8%)
Multifocal usual type VIN	39 (9.6%)
Mixed dystrophy	88 (21.6%)
Histological positive margin	92 (22.5%)
Thickness (mm)	3 [2–5]
Lymphovascular space invasion	23 (5.7)
**Vulvectomy**	
Partial	55 (13.5%)
Anterior	70 (17.2%)
Lateral	52 (12.7%)
Posterior	40 (9.8%)
Total	184 (45.1%)
**Lymphadenectomy**	355 (87%)
Unilateral dissection	77 (18.9%)
Bilateral dissection	278 (68.1%)
**Short-term complications**	
Infection	66 (16.2%)
Wound dehiscence	84 (20.6%)
Lymphocele	44 (10.8%)
Phlebitis	14 (3.4%)

Quantitative variables: median (interquartile range); qualitative variables: N (%).

**Table 2 jcm-09-00250-t002:** Predictive factors of lymph node involvement in univariate and multivariate analysis (*n* = 337 patients).

	Lymph Node Involvement		Univariate Analysis		Multivariate Analysis **	
	No (*n* = 211)	Yes (*n* = 126)	OR [95% CI]	*p **	OR [95% CI]	*p **
**Age, years**	71.52 [56.5–78.4]	71.93 [61.2–80.5]	–	0.38	–	
**Body mass index, Kg/m^2^**	24 [21.3–27.9]	24 [20.8–28.3]	–	0.8	–	
**Lichen sclerosis**	53 (25.1%)	49 (38.9%)	1.89 [1.15–3.13]	0.009	2.3 [1.2–4.28]	0.01
**Usual type VIN**	74 (35.1%)	30 (23.8%)	0.38 [0.33–0.98]	0.04	–	–
**FIGO**						
I	113 (53.6%)	0				
II	89 (42.2%)	0				
III	0	92 (73.0%)				
IV	8 (3.8%%)	33 (26.2%)				
**Local extension**						
Tumor confined to vulva						
size < 2 cm	110 (52.1%)	24 (19.0%)	0.22 [0.12–0.37]	<0.001	0.3 [0.02–5.2]	0.4
Tumor confined to vulva						
size > 2 cm	88 (41.7%)	76 (60.3%)	2.11 [1.32–3.41]	0.001	0.9 [0.1–15.3]	0.9
Tumor size > 2 cm	10 (4.7%)	25 (19.8%)	6.02 [2.86–12.66]	0.001	3.2 [0.2–62.3]	0.4
and/or extension to vagina. urethra or anus						
**Thickness (mm) ‡**	3.0 [2.0–5.0]	4.0 [1.5–6]	–	<0.01	1.1 [1.0–1.3]	0.1
**Positive margins**	46 (21.8%)	23 (18.3%)	0.81 [0.44–1.45]	<0.01	1.1 [0.5–2.3]	0.8
**Lymphovascular space invasion**	5 (2.4%)	14 (11.1%)	5.16 [1.70–18.81]	<0.01	5.6 [1.7–18.6]	0.05

OR, odds ratio; CI, confidence interval. Quantitative variables: median (interquartile range). Qualitative variables: N (%). * *p* value by Kruskal–Wallis, Chi2 or Fisher test, as appropriate. Odds ratios and 95% confidence intervals were estimated using univariate logistic regression. ** Multivariate analysis was adjusted for the 7 variables listed in the table. ‡ Odds ratio and confidence interval is expressed for 5 mm increase.

**Table 3 jcm-09-00250-t003:** Clinical characteristics correlated with lichen sclerosis.

	Lichen Sclerosis		
	No (*n* = 295)	Yes (*n* = 113)	*p **
**Age, years**	66.0	71.6	0.005
**Body mass index, Kg/m^2^**	24 [21.1–28.0]	24 [21.4–28.0]	
**Anterior localization**	174 (62.8%)	84 (75.2%)	0.017
**FIGO classification**			
I	106 (38.3%)	27 (22.1%)	0.009
II	63 (22.7%)	29 (25.9)	0.51
III	56 (20.2%)	36 (32.1%)	0.01
IV	33 (11.0%)	15 (13.4%)	0.73
**Local extension**			
Tumor confined to vulva, size < 2 cm	128 (46.2%)	34 (30.4%)	0.004
Tumor confined to vulva, size > 2 cm	114 (41.2%)	68 (60.7%)	0.0005
Tumor size > 2 cm and/or local extension to vagina, urethra or anus	31 (11.2%)	9 (8.0)	0.69
**Positive margins**	15 (13.4%)	74 (26.7%)	0.003
**Thickness**	3	4	<0.0001
**Lymphovascular space invasion**	16 (5.8%)	5 (4.5%)	0.8
**Positive lymph node**	77 (27.8%)	49 (43.8%)	0.009

Quantitative variables: median (interquartile range). Qualitative variables: N (%). * *p* value by Kruskal–Wallis, Chi2 or Fisher test, as appropriate.

**Table 4 jcm-09-00250-t004:** Classification of epithelial disorders associated with invasive vulvar carcinoma [8,11,14,15].

	Usual VIN	Differentiated VIN
Risk factors/associated dermatosis	HPV induced	Lichen
Type of squamous cell carcinoma	Warty or basaloid	Keratinized
Clinical types	Unifocal/multifocal	Generally unifocal
Progression to invasive carcinoma	5.7%	33%
Immunohistochemistry	p16+, p53-	p53+ (85%)

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
