# Peer review of "Associated Lichen Sclerosis Increases the Risk of Lymph Node Metastases of Vulvar Cancer"

_jcm, 2020, doi:10.3390/jcm9010250_

Round 1

Reviewer 1 Report

It is a high quality paper with  significant impact on clinical practice. Lichen sclerosus (LS) detection may be important in prevention and therapy of vulvar cancer since LS increases the risk of progression to invasive squamosus cell carcinoma (SCC) in a short time. SCCs constitute the majority of vulvar cancers. Therefore, the data are so important to improve poor quality of life of the patient with SCC.

Author Response

All the authors would like to sincerely thank the reviewer for his comments. 

Like him, we feel our results are important and might help clinicians decision making.

Reviewer 2 Report

This study showed that LS surrounding vulvar cancer is an independent prognostic factor, with local extension and LVSI.

In method: Did both centers perform sentinel node mapping with same protocol since 2002? If yes, please describe the method in more detail.

Did both centers receive central view for pathological confirmation by pathologist?

What treatments did patients receive after surgery and were they performed according to the same protocol? 

In result: Is there any data showing association with other immunostaining results such as p53? 

Is there any difference between the data at which preoperative biopsy is found and the rate at which LS was found after the definite surgery? This is because lymph node metastasis is sufficiently predictable for imaging.

In conclusion: "A biopsy should be performed to identify invasive lesion as soon as a suspect area is observed as LS increases the risk of progression to invasive SCC in the short term" : As a result of this study, it is difficult to whether LS can be used as a marker for predicting invasiveness.
